**Data Availability Statement:** Assembly-assembly alignment data displayed in the Comparative Genome Viewer are publicly available in GFF3 and

METHODS AND RESOURCES

# The NCBI Comparative Genome Viewer (CGV) is an interactive visualization tool for the analysis of whole-genome eukaryotic alignments

**Sanjida H. Rangwala** ⬤*, **Dmitry V. Rudnev, Victor V. Ananiev, Dong-Ha Oh**⬤,
**Andrea Asztalos, Barrett Benica, Evgeny A. Borodin, Nathan Bouk, Vladislav I. Evgeniev,**
**Vamsi K. Kodali**⬤, **Vadim Lotov, Eyal Mozes, Marina V. Omelchenko, Sofya Savkina,**
**Ekaterina Sukharnikov, Joël Virothaisakun, Terence D. Murphy, Kim D. Pruitt,**
**Valerie A. Schneider**

National Center for Biotechnology Information, National Library of Medicine, National Institutes of Health
(NIH), Bethesda, Maryland, United States of America

* rangwala@nih.gov

## Abstract

We report a new visualization tool for analysis of whole-genome assembly-assembly alignments, the Comparative Genome Viewer (CGV) (https://ncbi.nlm.nih.gov/genome/cgv/). CGV visualizes pairwise same-species and cross-species alignments provided by National Center for Biotechnology Information (NCBI) using assembly alignment algorithms developed by us and others. Researchers can examine large structural differences spanning chromosomes, such as inversions or translocations. Users can also navigate to regions of interest, where they can detect and analyze smaller-scale deletions and rearrangements within specific chromosome or gene regions. RefSeq or user-provided gene annotation is displayed where available. CGV currently provides approximately 800 alignments from over 350 animal, plant, and fungal species. CGV and related NCBI viewers are undergoing active development to further meet needs of the research community in comparative genome visualization.

## Introduction

### Comparative genome visualization

Comparative genomics leverages shared evolutionary histories among different species to answer basic biological questions and understand the causes of disease. As sequencing costs have dropped and assembly algorithms have improved, there has been tremendous growth in the number of high-quality genome assemblies available in public archives and the diversity in the organisms they represent. These data now make it possible to use comparative genomics approaches to explore more elements of biology and reveal the need for different types of analysis tools to support this exploration. The NIH Comparative Genomics Resource (CGR)

ASN.1 formats at https://ftp.ncbi.nlm.nih.gov/pub/remap/. Gene annotation can be accessed via NCBI Datasets command line interfaces at https://www.ncbi.nlm.nih.gov/datasets/docs/v2/download-and-install/.

**Funding:** This work was supported by the National Center for Biotechnology Information of the National Library of Medicine (NLM) at the National Institutes of Health (NIH). The funders had no role in study design, data collection and analysis, decision to publish, or preparation of the manuscript.

**Competing interests:** The authors have declared that no competing interests exist.

**Abbreviations:** CDS, coding sequence; CGR, Comparative Genomics Resource; CGV, Comparative Genome Viewer; GDV, Genome Data Viewer; IGV, Integrative Genomics Viewer; NCBI, National Center for Biotechnology Information; USWDS, United States Web Design System.

maximizes the impact of eukaryotic research organisms and their genomic data to biomedical research [1]. CGR facilitates reliable comparative genomics analyses for all eukaryotic organisms through community collaboration and a National Center for Biotechnology Information (NCBI) genomics toolkit. As part of CGR, we have created the Comparative Genome Viewer (CGV), a web-based visualization tool to facilitate comparative genomics research.

Graphical visualization of genomic data can illuminate relationships among different data types and highlight differences and anomalies; for example, areas of a genome that are depleted in gene annotation, or have unusually high repeat content, or are more variable between species. Interactive genome browsers have become particularly valuable in recent years in helping biologists navigate large sequence datasets and more easily find genomic locations of interest to their specific research question. These visualizations can display molecular data that can help resolve competing hypotheses and expose patterns that can spur additional research questions.

Linear genome browsers, such as the Genome Data Viewer (GDV) at NCBI [2], the UCSC Genome Browser [3], Integrative Genomics Viewer (IGV) [4], and JBrowse [5], visualize gene annotation, sequence variation, and other types of molecular data as "tracks" laid out in parallel and anchored on a single genome assembly. These browsers can display comparative genome data such as genome sequence alignments (e.g., UCSC's chain/net or NCBI assembly alignments), which can provide an indication of conservation in a discrete genomic region. However, linear genome browser tracks cannot show as easily whether a genome region has been rearranged (e.g., translocated to different chromosomes) in one genome relative to another one. Genome translocations and other types of breaks in syntenic blocks are more easily communicated using 2D viewers that can provide a whole-genome comparison of alignments of one genome relative to another.

Different types of 2D visualizations have been proposed to facilitate analysis of larger scale genome structural differences between 2 or more genomes. These visuals include 2D line graphs (also known as dotplots) [6,7], circular diagrams (i.e., Circos plots) [8], vertical genome maps (e.g., Ensembl's synteny view [9]), and linear genome browsers that stack one assembly on top of another [10–12]. Different types of visuals have advantages and disadvantages. Circos plots can show multiple datasets in one graphic but can be visually challenging to interpret and usually do not support views of sub-genomic regions. Dotplots can allow zooming to view chromosome or sub-chromosome regions but cannot easily or elegantly display gene or other annotation in the same visual. In order to better serve different research questions, some groups provide a choice of multiple different types of visuals for genome comparisons [13–15].

## Genome comparison data

Broadly, there are 2 types of whole-genome comparison data that can be displayed in a comparative genome visualization tool. The first type of data is locations of gene orthologs. Orthology is typically determined using a protein homology-based method (e.g., protein–protein BLAST) in consideration with local gene order conservation [16,17]. This type of data can lend itself to straightforward "beads on string" visualizations that allows researchers to easily determine how syntenic gene regions have evolved across different species [17–19].

The second type of comparison data is whole-genome assembly alignments (e.g., Mauve [20], LASTZ [21]), which are sequence-based and include both genic and intergenic regions. Whole-genome alignments can be much more complex than simple gene ortholog locations but have the advantage of including alignments in regulatory regions and other regions not annotated as genes.

Here, we introduce a new viewer tool at NCBI, the CGV, that is a key element of CGR. The main view of CGV takes the "stacked linear browser" approach—chromosomes from 2 assemblies are laid out horizontally with colored bands connecting regions of sequence alignment. Initial usability research with conceptual prototypes revealed that this type of visual was the easiest to interpret for scientists with a broad range of research expertise in genomics. We display whole-genome pairwise assembly-assembly alignments in CGV. These sequence-based alignments can be used to analyze gene synteny conservation but can also expose similarities in regions outside known genes, e.g., ultraconserved regions that may be involved in gene regulation. Because CGV is a web-based application, researchers do not need to install or configure software or generate their own comparison files before they can begin using it for their research. Below we describe some of the features of CGV and provide examples of how visualization in this tool can generate insights into genome structure and evolution.

## Results

### Overview of CGV

We developed a web application, the CGV (https://ncbi.nlm.nih.gov/genome/cgv/), to aid in comparing genome structures between 2 eukaryotic assemblies. CGV facilitates analyses of genome variation and evolution between different strains or species, as well as evaluation of assembly quality between older and newer assemblies from the same species.

Alignments are generated at NCBI using BLAST [22] or LASTZ-based algorithms [21] or imported from the UCSC Genomics Institute (https://hgdownload.soe.ucsc.edu/downloads.html) and other research groups (e.g., T2T/HPRC, https://humanpangenome.org/). Shorter alignments are merged where possible; however, because of repeats and gaps, even very similar genomic regions may be broken down into multiple alignment segments. More closely related genomes will provide more contiguous alignments, while more distant species may align only to short highly conserved regions. In addition, while we are often able to provide alignments for polyploid genomes, it is more difficult to distinguish orthologs (identity by descent) from homeologues (identity by duplication) for species pairs with more recent whole-genome duplications. CGV is therefore more suited to analyzing alignments for polyploids with more distinct sub-genomes, e.g., allopolyploids or older autopolyploids (S1 Appendix). Refer to **Materials and methods** for more details on how we generate whole-genome assembly alignments and load them into the viewer.

The CGV home page provides a menu where users can select from available species and assembly combinations (Fig 1A). We add new whole-genome alignments as high-profile assemblies become available and in response to requests from the scientific community. As of February 2024, we provided a selection of about 800 alignments from over 350 eukaryotic species (Fig 1B). Whole-genome sequence alignments between more distantly related species may be sparse or low-quality with limited analytical utility; therefore, most of the alignments we offer are between assemblies of the same species or more closely related species within the same class or order.

CGV's main view (the "ideogram view") displays pairwise alignments as colored connectors linking the chromosomes in the 2 assemblies (Fig 1C). The view is filtered by default to show only reciprocal best hits between assemblies in order to facilitate the analysis of orthologous genomic regions. Researchers can choose to show the non-best placed alignments to reveal additional closely related sequence duplications or ancestral homologues (Fig 1F and S1 Appendix). Users of CGV can also filter alignments in view by size (e.g., to only show large alignment blocks) or by orientation (e.g., to only show regions that have undergone a potential

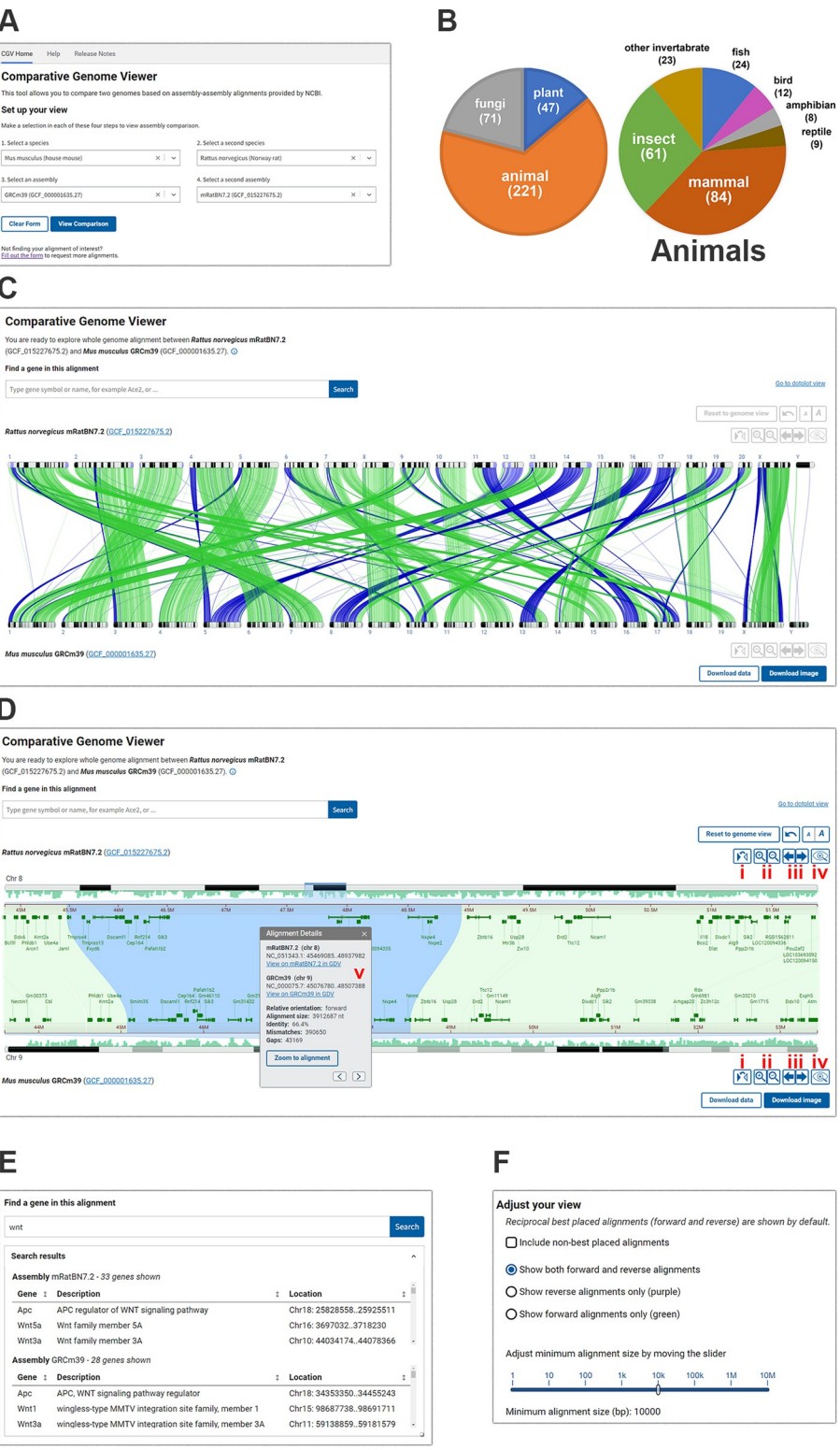

**Fig 1. Overview of CGV.** (A) CGV selection menu. (B) Taxonomic distribution of species represented by alignments in CGV. Numbers in the pie charts are current as of February 5, 2024 (C) CGV ideogram view of whole-genome assembly alignment. Buttons in the lower right provide download access for complete whole-genome alignment data or an SVG image of the current alignment view. The view can be recreated at https://www.ncbi.nlm.nih.gov/genome/cgv/browse/GCF_015227675.2/GCF_000001635.27/27835/10116 (D) CGV zoomed to a chromosome-by-

chromosome view, with an information panel shown. This panel can be viewed by clicking to select an alignment segment. (i) Flip orientation; (ii) zoom in/out; (iii) pan left/right; (iv) view assembly in GDV; (v) information panel. The view can be recreated at https://www.ncbi.nlm.nih.gov/genome/cgv/browse/GCF_015227675.2/GCF_000001635. 27/27835/10116#NC_051343.1:44657546-50471623/NC_000075.7:41365809-52285446/size=10000. (E) CGV search interface and sample search results. (F) Adjust Your View configure options for the ideogram view. Webpage source: National Library of Medicine. CGV, Comparative Genome Viewer; GDV, Genome Data Viewer.

inversion). The complete whole-genome alignment data in GFF3 and human-readable formats like XLSX can be downloaded from the viewer for a researcher's own use.

Users can click to select a chromosome from each assembly to zoom to the alignments for the selected chromosome. They can navigate further within this chromosome comparison using the zoom in/out and pan buttons or by pinch-zoom or drag to pan. Users can zoom directly to a particular region of a chromosome by dragging their cursor over the coordinate ruler or the ideogram for either assembly. Double-clicking on a selected alignment segment will synchronously zoom both the top and bottom assembly on the aligned coordinates so that they are stacked on top of one another (Fig 1D).

Where available, RefSeq or assembly-submitter provided gene annotation is displayed on the chromosomes (Fig 1D). Similarities in gene order denote regions of synteny, while discrepancies can point to evolutionarily or biologically significant differences. Differences may also result from assembly errors, particularly if evaluating different assemblies from the same species or strain. Researchers can use the search feature in CGV to find their gene of interest by name or keyword, and subsequently navigate to the location of the gene in the viewer (Fig 1E). If the gene region is aligned, the viewer will simultaneously navigate to the aligned location, which may contain the gene's known or putative ortholog on the second assembly. The "flip" button allows the user to reverse one chromosome to see inverted alignments displayed in the same relative orientation, which may aid in the detection of discrepancies in gene annotation in regions that are locally syntenic between the 2 assemblies. Once a user has completed their analysis of a region of interest, they can export the image as an SVG to adapt for use in publications and presentations.

Users can click on an alignment segment to show an information panel (Fig 1D). This panel reports the chromosome scaffold accession and sequence coordinates of the alignment on each assembly, as well as the percent identity, number of gaps and mismatches, and alignment length. While the ideogram view in CGV does not display specific nucleotide bases, users can open another panel from the right-click menu that shows the alignment sequence. They can also download the alignment FASTA file of a particular alignment segment for downstream analysis, such as BLAST search or primer design. Researchers can also navigate from CGV to NCBI's genome browser, the GDV [2]. GDV can display the assembly-alignment data viewed in CGV as a linear track alongside additional data mapped onto a genome assembly, such as detailed transcript and CDS annotation, repeats, GC content, variation data, or user-provided annotations. Zooming to a location within GDV can reveal differences in nucleotide sequence or gene exon or CDS annotation between the 2 assemblies.

In addition to the main ideogram-based view, the Comparative Genome Viewer also provides a 2D dotplot view of the pairwise genome alignment (Fig 2A). The dotplot shows aligned sequence locations in one assembly on the X-axis plotted against aligned locations on the second assembly on the Y-axis. Alignments in the reverse orientation are plotted with an opposite slope and in a different color (purple) than alignments in the same orientation (green), making it easier to identify inversions and inverted translocations. The CGV dotplot shows both reciprocal best-placed and non-best placed alignments. As a result, compared to the ideogram view, this plot may more easily expose differences in copy number between 2 assemblies, such as

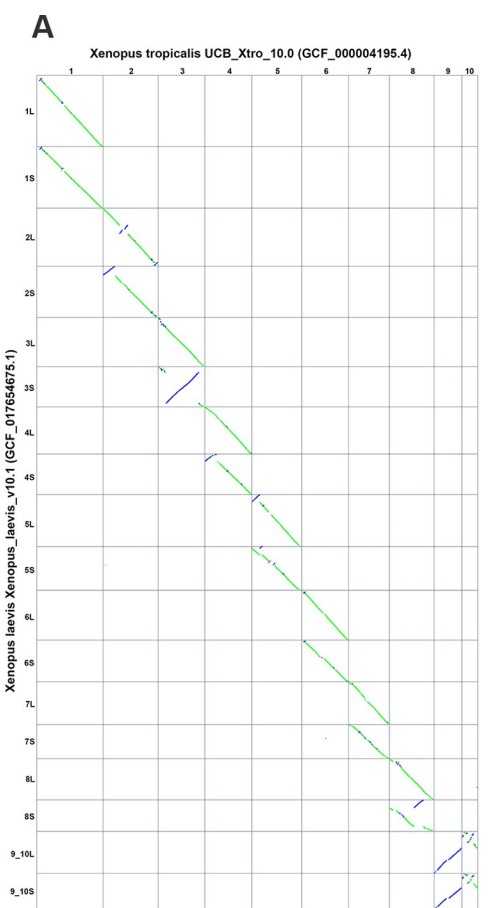

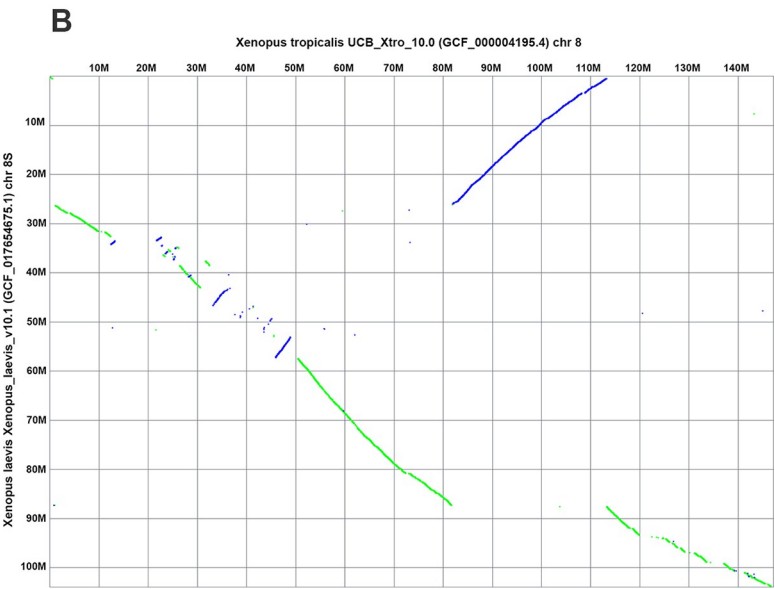

**Fig 2. CGV dotplot view of *Xenopus laevis* and *Xenopus tropicalis* alignment.** (A) Full genome dotplot. (B) Dotplot of chromosome 8 of *Xenopus tropicalis* vs. chromosome 8S of *Xenopus laevis*. These views can be recreated at https://ncbi.nlm.nih.gov/genome/cgv/plot/GCF_000004195.4/GCF_017654675.1/38475/8355. CGV, Comparative Genome Viewer.

segmental duplications or differences in genome or chromosome ploidy. Users can select and zoom to a view showing the comparison between a pair of chromosomes in the whole-genome plot (i.e., a "cell" in the plot) (Fig 2B). Once a researcher has discovered a chromosome pair of interest in the dotplot, they can navigate back to the ideogram view to conduct even more detailed analysis, including examining gene annotation and investigating short alignment segments that were beyond the resolution of the dotplot.

## Analysis using CGV: Conservation of linkage groups with local rearrangement of synteny

CGV can aid in detecting unusual patterns in genome evolution in different taxa. Researchers had previously observed that genomes from *Drosophila* species conserve gene content within linkage groups, known as Muller elements, corresponding to chromosomes or large sub-chromosome regions. Within these Muller elements, gene order can be reshuffled extensively in one species relative to another [23]. For alignment between *D. albomicans* versus *D. melanogaster*, the CGV ideogram view shows restriction of alignment from each chromosome in one genome to a particular chromosome or chromosome region (i.e., linkage group) in the other genome (Fig 3A). However, within a chromosome–chromosome pair, sequence alignment is broken into many small fragments whose relative order is not conserved. This fragmentation of alignment is more clearly visible in the CGV dotplot, which shows that pairwise alignments are restricted to a single chromosome pair, but appear in a scattered pattern, suggesting that the sequence and gene order has been extensively rearranged within chromosomes (Fig 3B).

   We observed a similar pattern to *Drosophila* when comparing some genomes from different starfish species using CGV. When looking at pairwise alignments between *Asterias rubens*, *Patiria pectinifera*, and *Luida sarsii* species, sequences from a chromosome from 1 genome mainly align to a single other chromosome in the other species. However, within a pairwise chromosome alignment, the sequence order is rearranged, resulting in a scatter pattern in the dotplot (Fig 3C and 3E). The ideogram view can show the alignment fragmentation and rearrangement in more granular detail (Fig 3D). We also noted that some starfish pairs show more conservation of location synteny [24] (Fig 3F), consistent with measured sequence distance (Mash distance) based on shared k-mers [25] (Mash distance between *Plazaster borealis* and *Pisaster ochraceus* is 0.128 compared to other starfish pairs with a Mash distance >0.3).

   Bhutkar and colleagues [23] speculated that the need to keep certain genes in the same regulatory environment may result in conservation of genes within linkage groups even in the absence of selective pressure to maintain the gene order. More recently, conservation of macrosynteny with extensive small-scale sequence rearrangement was reported in comparisons between other invertebrate species, such as cephalopods, cnidarians, jellies, and sponges [18,26,27]. These rearrangements were used to parse the phylogenetic relationships within this clade. We demonstrate here that CGV can aid researchers in detecting and analyzing this phenomenon in starfish and other evolutionarily varied taxa.

## Analysis using CGV: Detection of gene family expansions between related genomes

CGV can uncover potential copy number differences in segmental gene families. These differences may appear as gaps in the alignment in otherwise syntenic gene regions. Segmental insertions or deletions may be too small to be apparent on the whole-genome or whole-chromosome alignment but can be detected when searching and navigating to a gene of interest.

   Initial analysis of the complete human telomere-to-telomere CHM13 (T2T-CHM13) genome indicated an expansion of amylase genes on chromosome 1 in the T2T assembly

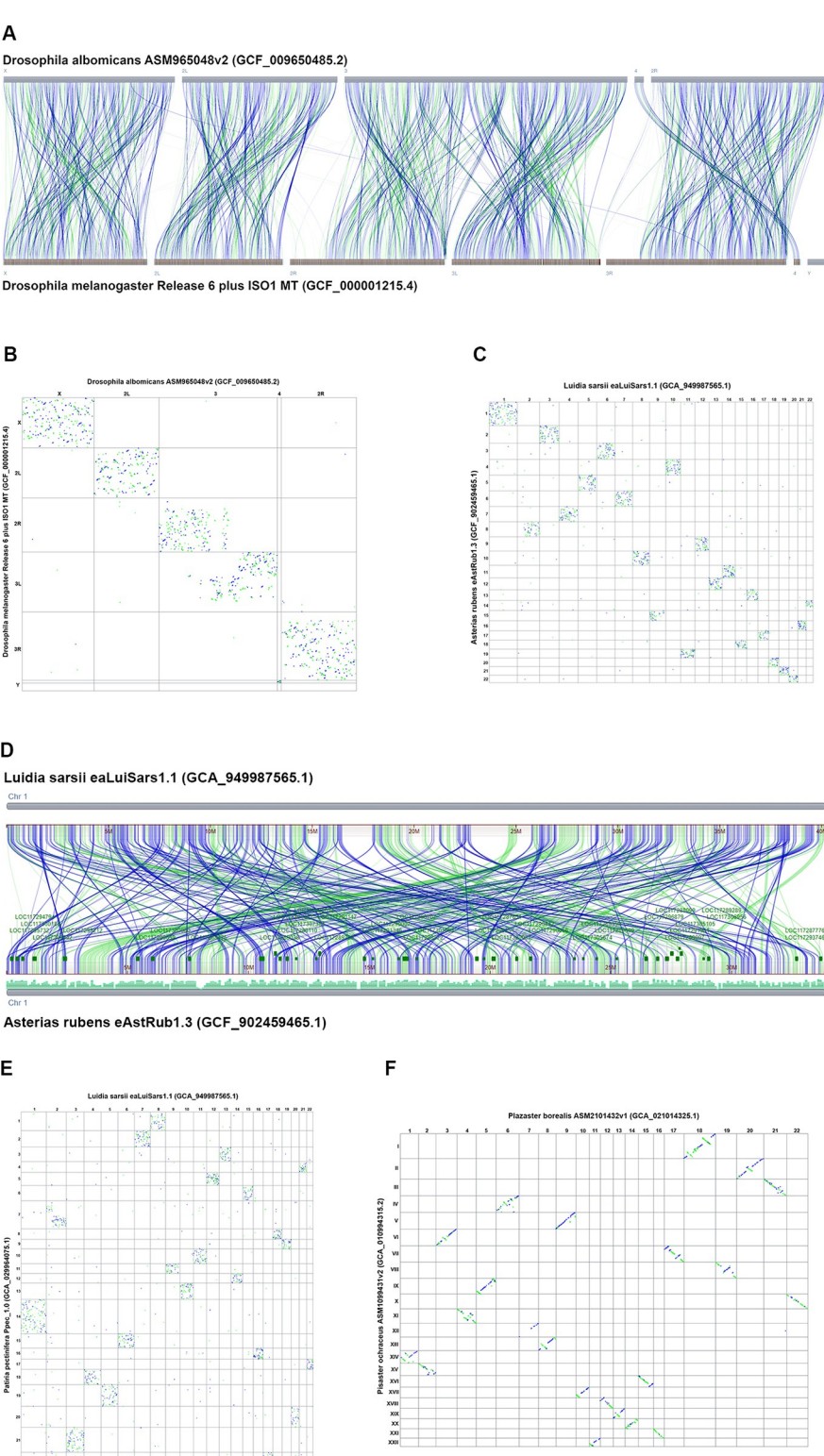

**Fig 3. CGV shows conservation of linkage groups in the absence of conservation of gene order.** (A) CGV ideogram view of alignment between *Drosophila albomicans* and *Drosophila melanogaster* genomes. Alignments are restricted to a single chromosome or chromosome region. The view can be recreated at https://www.ncbi.nlm.nih.gov/genome/cgv/browse/GCF_009650485.2/GCF_000001215.4/40865/7291. (B) CGV dotplot view of alignment between *Drosophila albomicans* and *Drosophila melanogaster* demonstrates that sequence order is "scrambled" within linkage groups, as

demonstrated by a scatter pattern indicating many short rearranged alignments. The view can be recreated at https://www.ncbi.nlm.nih.gov/genome/cgv/plot/GCF_009650485.2/GCF_000001215.4/40865/7291. (C) CGV dotplot view of alignment between starfish species *Luida sarsii* and *Asteria rubens* with similar scatter pattern to *Drosophila* alignments. The view can be recreated at https://www.ncbi.nlm.nih.gov/genome/cgv/plot/GCA_949987565.1/GCF_902459465.1/41045/2723838. (D) CGV ideogram view of alignment between chromosome 1 of *Luida sarsii* and chromosome 1 of *Asteria rubens*. These chromosomes align to each other across their length, but the alignment is broken into multiple short segments which are extensively rearranged. The view can be recreated at https://www.ncbi.nlm.nih.gov/genome/cgv/browse/GCA_949987565.1/GCF_902459465.1/41045/2723838#OX465101.1/NC_047062.1/size=1,firstpass=0. (E) CGV dotplot view of alignment between starfish species *Luida sarsii* and *Patiria pectinifera* with similar scatter pattern to *Drosophila* alignments. The view can be recreated at https://www.ncbi.nlm.nih.gov/genome/cgv/plot/GCA_949987565.1/GCA_029964075.1/41165/7594. (F) CGV dotplot view of alignment between starfish species *Plazaster borealis* and *Pisaster ochraceus*. Alignments show less scatter and more of a diagonal slope, indicating more conservation of sequence order between these 2 species' genomes. The view can be recreated at https://www.ncbi.nlm.nih.gov/genome/cgv/plot/GCA_021014325.1/GCA_010994315.2/41175/466999. CGV, Comparative Genome Viewer.

compared to the human GRCh38 reference assembly [28,29]. The whole-genome alignment between GRCh38.p14 and T2T-CHM13v2.0 generated by the T2T/HPRC consortium [29] is available to analyze in CGV and validates this initial finding. A search for "alpha amylase" in CGV finds 4 matches in the GRCh38.p14 assembly and 10 matches in the T2T-CHM13v2.0 assembly (Fig 4A). Navigating to the *AMY1A* gene on chromosome 1 reveals a nearby sequence segment in the T2T-CHM13 assembly that is not aligned to GRCh38 (Fig 4B). This region in the T2T-CHM13 assembly contains numerous annotated loci that lack official gene nomenclature (i.e., named only as "LOC" followed by the numerical locus identifier ID); 6 of these loci are described as "alpha-amylase" gene family members (Fig 4B). Therefore, there are at least 6 additional alpha-amylase genes in the T2T-CHM13 genome compared to the GRCh38 reference assembly. It is possible that the copy number of this gene is variable in humans; it is also possible that the GRCh38reference genome represents fewer than the typical number of gene copies.

The T2T/HPRC alignment described above contains only reciprocal best placed alignments; therefore, it does not include alignments corresponding to the additional copies of alpha amylase in the T2T-CHM13v2.0 assembly. As a result, you will not be able to see alignments to these paralogs, even when the option to "show non-best placed alignments" is checked on. S1 Appendix (B–E) provides additional examples of genomic regions visualized in CGV that feature local gene copy number expansions in one genome relative to another. These alignments were generated by NCBI (see **Materials and methods**; S1 Fig) and, as a result, include non-best placed alignments in the alignment data. Turning on the option to "show non-best placed alignments" reveals that single copy genes in the human genome (e.g., *BMPR2*, *EIF4A3*, or *EYS)* are present in multiple copies in the corresponding genomic region of other primates (S1 Appendix B–E). These primate assemblies are likely to be high quality, so that the copy number differences observed are likely to be real. In situations where one or both genome assemblies are of lower quality, differences in copy number observed in CGV may reflect assembly or annotation errors.

## Analysis using CGV: Possible gene translocation between 2 dog assemblies

For closely related strains or species, CGV can help uncover and validate structural anomalies, such as where gene order synteny has been disrupted. Visual inspection of the whole-genome CGV ideogram view of alignment between 2 dog genomes—the Great Dane Zoey (UMICH_-Zoey_3.1) and the boxer Tasha (Dog10K_Boxer_Tasha)—indicated a region that aligned to chromosome 2 in the Zoey assembly and chromosome 25 in the Tasha assembly (Fig 4C and 4D). This region contains the *MALRD1* gene in the Zoey assembly, which is shown to align to

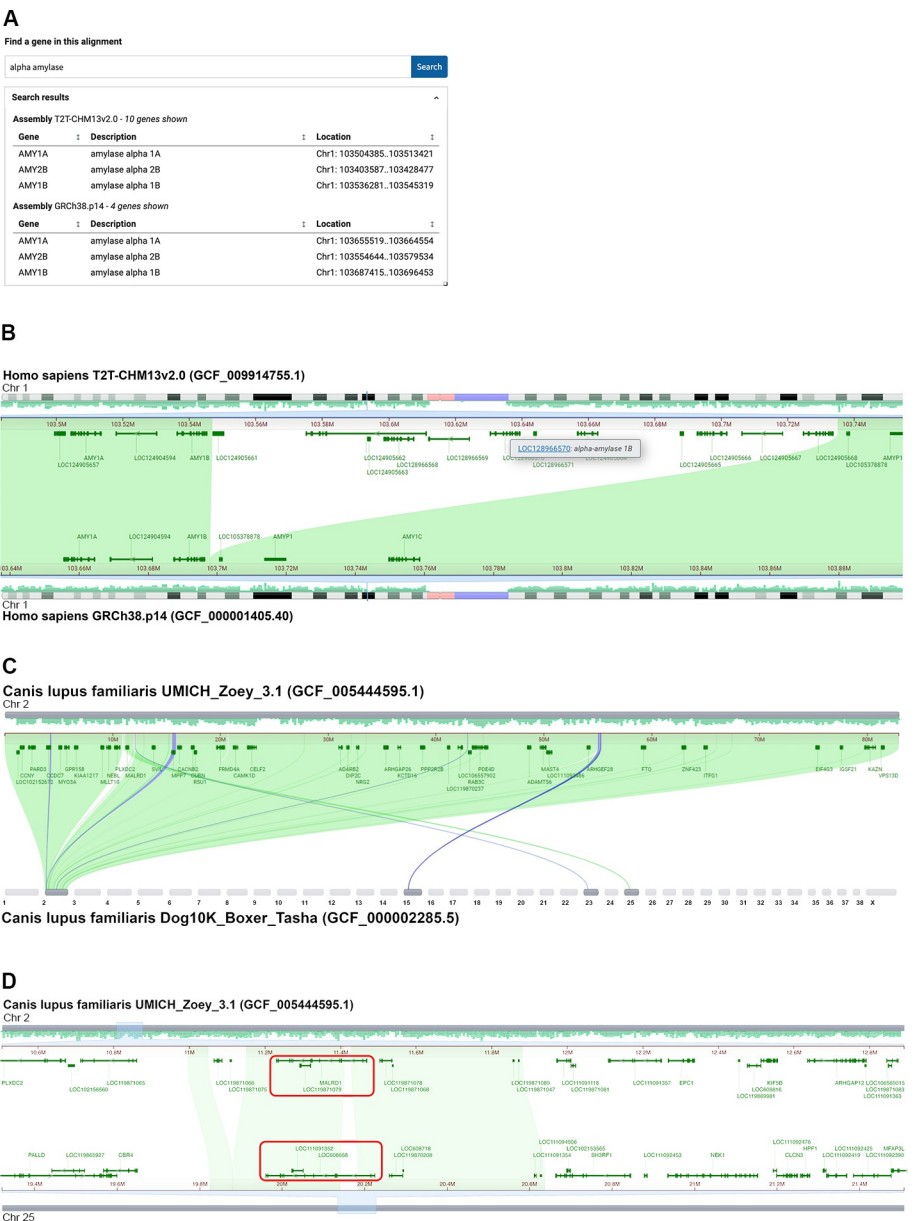

**Fig 4. CGV can help uncover gene duplications and rearrangements in closely related genomes.** (A) Gene search of a T2T/HPRC-generated alignment between 2 human assemblies in CGV finds an alpha-amylase gene cluster on chromosome 1 containing 10 copies in the T2T-CHM113v2.0 assembly and 4 copies in the GRCh38.p14 assembly. (B) CGV view showing that T2T-CHM13v2.0 contains an insertion relative to GRCh38.p14, which appears as an unaligned region on chromosome 1. This insertion contains additional alpha-amylase gene copies. An information panel (tooltip) indicates one of these additional family members. These tooltips appear when the user hovers their cursor over the gene annotation or gene label. The view can be recreated at https://www.ncbi.nlm.nih.gov/genome/cgv/browse/GCF_009914755.1/GCF_000001405.40/23025/9606#NC_060925.1:103415704-103764412/NC_000001.11:103566852-103915505/size=1000,firstpass=0. (C) CGV view showing that chromosome 2 of *Canis lupus familiaris* (dog) UMICH_Zoey_3.1 aligns to chromosomes 2, 15, 23, and 25 of Dog10K_Boxer_Tasha. The view can be recreated at https://ncbi.nlm.nih.gov/genome/cgv/browse/GCF_005444595.1/GCF_000002285.5/17685/9615#NC_049262.1:6542815-78714085//size=10000. (D) UMICH_Zoey_3.1 assembly chromosome 2 alignment to Dog10K_Boxer_Tasha chromosome 25 contains the *MALRD1* gene, which is annotated as *LOC608668* in the Tasha assembly (boxed in red). Gene synteny is not conserved outside of the region of this gene. Webpage source: National Library of Medicine. CGV, Comparative Genome Viewer.

a *MALRD1* homolog annotated as *LOC608668* in Tasha (Fig 4D). CGV alignments indicate that *LOC608668* is likely the Tasha *MALRD1* gene; there are no better alignments to *MALRD1* detected in CGV or by an independent BLAST search of the Tasha genome.

Zooming out in the aligned region of *MALRD1* indicates that gene synteny is not conserved outside of this gene r (Fig 4D). It appears that this gene has been translocated from chromosome 2 on the Zoey assembly to chromosome 25 on Tasha. It is also possible that the Tasha genome may have been misassembled in this region, and the *MALRD1* gene sequence is properly situated on chromosome 2 within the otherwise conserved syntenic block. A researcher would need to further examine the quality of the Tasha assembly in this region to distinguish these possibilities, for example, by examining the sequencing reads for the Tasha assembly or viewing an HiC map of assembly structure.

If this anomaly represents a true difference between the 2 genomes, it could prove biologically significant. The translocation of *MALRD1* may have placed it in a different gene regulatory environment in the Tasha genome, which could possibly result in different levels or patterns of gene expression. The human ortholog of *MALRD1* was shown to be involved in bile acid metabolism [30]. This gene region was also genetically linked to Alzheimer's disease [31]. Therefore, if valid and not assembly artifacts, differences like these could provide insight into human health.

## Discussion

We describe here a new visualization tool for eukaryotic assembly-assembly alignments, the CGV. We developed this web application with a view toward serving both expert genome scientists as well as organismal biologists, students, and educators. Users of CGV do not need to generate their own alignments or configure the software using command line tools. Instead, they can select from our menu of available alignments, access a view immediately in a web application, and start their analysis. We are continuing to add new alignments regularly and invite researchers to contact us if assemblies or organisms of interest are missing. We continue to do periodic outreach to the community to help us improve our visual interfaces so that they are simple, intuitive, and accessible.

CGV exclusively displays whole-genome sequence alignments provided by NCBI; users cannot currently upload their own alignment data or choose assemblies to align in real time. There are both technical and scientific considerations to allowing researchers to select and align assemblies automatically themselves. Currently, whole-genome assembly-assembly alignments take several hours to days, using up to 1,000 CPU processing hours per pairwise alignment of larger genomes, such as those for mammalian or plant assemblies. Moreover, whole-genome alignments are difficult to generate past a certain genetic distance (i.e., Mash > 0.3). Alignments between more distant species may be of limited research value as they will likely have sparse and short segments that may correspond only to the most highly conserved coding sequence (CDS) (Fig 5 and S1 Data). Some more closely related genomes may also be inappropriate for CGV's alignment algorithm. For example, our alignments may not accurately distinguish sub-genomes for genome pairs with recent whole-genome duplications. We also cannot accurately align assembly pairs where there are large differences in ploidy levels and genome sizes. We suggest protein similarity or gene orthology-based alignments as more appropriate for comparison between genomes that are not appropriate for CGV's whole-genome sequence analysis. At present, we manually vet requested alignments to make sure that requested assemblies are complete, of high quality (e.g., high scaffold N50 or BUSCO scores [32]), and at a reasonable evolutionary distance. This review ensures that alignments will be useful to both the original requester and others in the research community.

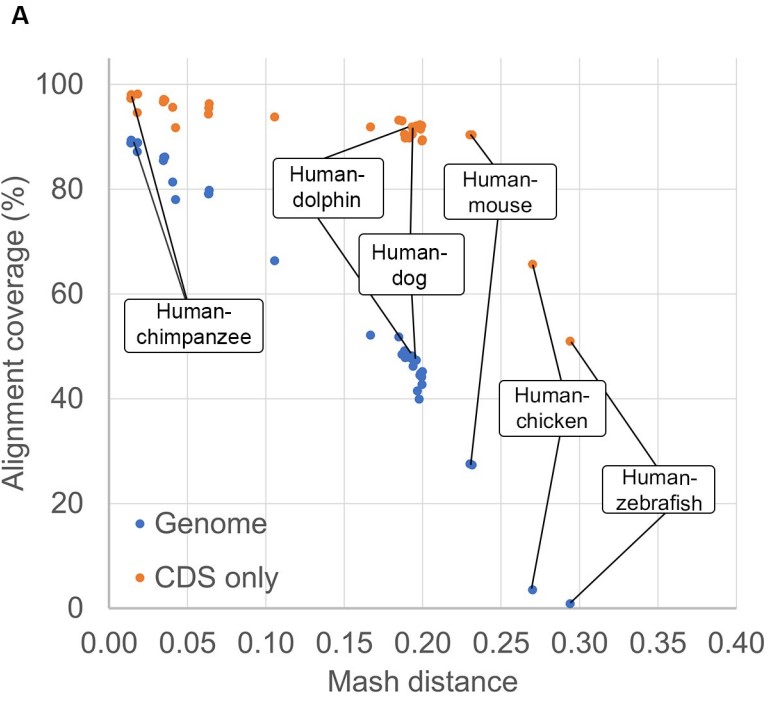

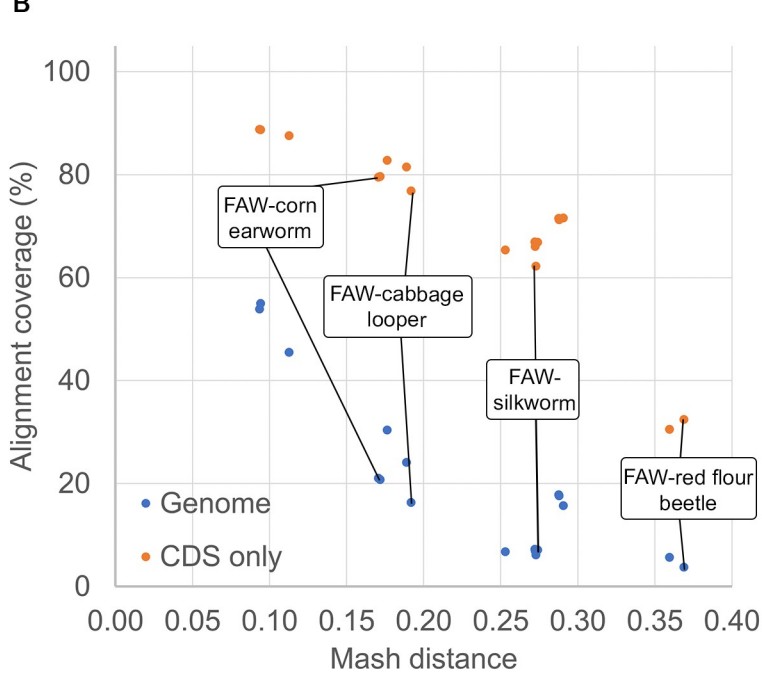

**Fig 5. Genome and CDS coverage of assembly-assembly alignments relative to Mash distance.** Percentages of total target genome or CDS nucleotides covered by ungapped alignments are plotted against the Mash distance between the pair of genomes. (A) Alignments between the human GRCh38.p14 assembly and other vertebrates. (B) Alignments of fall army worm (FAW, *Spodoptera frugiperda*, a major insect agricultural pest) and related insects. At lower Mash distances, the whole-genome alignments cover most of the genome and CDS. At Mash distances greater than 0.25 or 0.3, the alignment covers less than 20% of the genome overall, and between 30% and 60% of the CDS. Refer to S1 Data for the data used to populate these graphs. CDS, coding sequence.

Many research questions in comparative biology may be best answered by simultaneously visualizing alignments among more than 2 assemblies. We are exploring user needs for multi-genome alignment visualization in order to better support this analysis at NCBI. Our lessons from developing CGV will prove valuable in this upcoming initiative.

## Materials and methods

### Preparation of assembly-assembly alignments

Genome assemblies are aligned using a two-phase pipeline first described in Steinberg and colleagues [33], with adaptations for cross-species alignments. The pipeline is engineered in a custom database-driven workflow engine ("gpipe") that executes programs written in C++; the extensive internal database and C++ library dependencies preclude distribution of code for compilation or use outside of NCBI without extensive re-engineering. Here, we provide a detailed description of the processing. In the first phase, initial alignments are generated using BLAST [22] or LASTZ [21], or imported from a third-party source such as UCSC [34]. In the second phase, alignments are merged and ranked to distinguish reciprocal-best alignments from additional alignments that are locally best on one assembly but not the other. The resulting alignment set is omnidirectional and can be used to project information from query to subject assembly or vice versa.

In the first phase, for both BLAST and LASTZ, repetitive sequences present in the query and target assemblies are soft-masked using WindowMasker [35]. The default parameters are usually suitable for aligning assembly pairs within the same species. However, more aggressive masking is required when aligning cross-species assemblies. The masking rate is adjusted with the parameter `t_thres_pct` set to 99.5 (default, for BLAST same-species), 98.5 (for BLAST cross-species), or 97.5 (LASTZ cross-species), or lower for some genomes with extensive and diverse repeat composition. The 97.5 to 98.5 values typically result in a masking percentage similar to RepeatMasker (http://www.repeatmasker.org) with a species-specific repeat library, with the advantage of not needing to define repeat models beforehand.

Genome assemblies are aligned using either BLAST or LASTZ. The selection of the aligner and specific parameters depends on the level of similarity between the assemblies. We use Mash [25] to compute the approximate distance between 2 assemblies (Fig 5 and S1 Data). BLAST is employed for aligning pairs of assemblies belonging to the same species, as well as cross-species assembly pairs with a Mash distance less than 0.1. An exemplar BLAST command is:

```
blastn -evalue 0.0001 -gapextend 1 -gapopen 2 -max_target_
seqs 250 -soft_masking true -task megablast -window_size 150
-word_size 28
```

A BLAST `word_size` of 28 is used for pairs of assemblies with Mash distances below 0.05, such as human and orangutan, while a `word_size` of 16 is used to enhance sensitivity for more distant cross-species pairs with Mash distances ranging from 0.05 to 0.1, such as human and rhesus macaque.

Assembly pairs with Mash distances exceeding 0.1, such as human and mouse assemblies, are aligned using LASTZ. The `make_lastz_chains` pipeline [21] is employed to generate alignments between query and target assemblies in UCSC chain format. The default parameters are often adequate to produce satisfactory alignments for many assembly pairs, though some distant assembly pairs (e.g., Mexican tetra-medaka) warrant changes such as the use of a different substitution matrix (`BLASTZ_Q=HoxD55`).

Alignments generated with the `make_lastz_chains` pipeline or precomputed alignments imported from UCSC are in chain format. The UCSC chainNet pipeline [36] is run on

query x target and target x query alignment sets separately so that the alignments are "flattened" in a way that the reference sequences are covered only once by the alignments in each set, and the 2 chainNet outputs are concatenated.

In the second phase, alignments are converted to NCBI ASN.1 format and processed further for the NCBI's CGV and GDV browsers. In this phase, the full set of BLAST or LASTZ-derived alignments is processed to merge neighboring alignments and split and rank overlapping alignments to identify a subset of best alignments (S1A Fig). Merging is accomplished on a sequence-pair-by-sequence-pair basis, and ranking is accomplished globally for the assembly pair. The process is designed to find a dominant diagonal among a set of potentially conflicting alignments.

Merging involves the following steps. First, when applicable, alignments based on common underlying sequence components of the assemblies (e.g., the same BAC component used in both human GRCh37 and GRCh38) are identified and merged into the longest and most consistent stretches possible. Second, adjacent alignments are merged if there are no conflicting alignments. Third, alignments are split on gaps using a default threshold of 50 bp (for the same or closer species) or 50 kb (for more distant species), or longer than 5% of the alignment length. Alignments are also split at any point where they intersect with overlapping alignments (S1A Fig). Duplicate or low-quality alignments fully contained within higher-quality alignments are dropped.

After merging and splitting, alignments are subsequently processed using a sorting and ranking algorithm (S1B Fig). Alignments are sorted based on a series of properties, including the use of common components, assembly level (alignment to chromosomes preferred over alignment to unplaced scaffolds), total sequence identity, and alignment length. The alignments are then scanned twice, once each on the query and subject sequence ranges, to sort out reciprocal best-placed (also referred to as "first pass" or reciprocity = 3) and non-best placed (also referred to as "second pass" or reciprocity = 1 or 2) alignment sets. Finally, all alignments in each reciprocity are merged again to stitch together adjacent alignments with no conflicting alignments into the longest representative stretches.

Assembly-assembly alignments are stored in an internal database available for rendering in NCBI's CGV and GDV browsers. The alignment data are publicly available in GFF3 and ASN.1 formats at https://ftp.ncbi.nlm.nih.gov/pub/remap/.

For display in CGV, assembly alignment batches are filtered to keep only alignments that contain chromosome scaffolds as both anchors and targets, since non-chromosomal scaffolds are not displayed in this viewer. Alignments are converted into a compact binary format designed to keep only the synteny data required for display. This preparation step is done by programs written in C++ and bash scripts that tie them together.

## Technical architecture of CGV

CGV operates on a two-tier model, with a front end implemented using HTML/JavaScript running in the user's web browser and a back end running at NCBI. The graphical rendering is done on the front end using modern WebGL technologies. The main advantage of this approach is speed and fluidity of the user interface since most of the alignment data needed to be rendered is sent to the front end at the initial load and there are no additional roundtrips to the server when the user interacts with the page (e.g., panning or zooming). Using front end graphical rendering also reduces the network traffic between NCBI and the end user, which makes CGV more responsive.

The back end of the CGV application resolves internal alignment identifiers to an alignment data file that the front end can use for generating graphical images. The back end is

implemented as an industry-standard gRPC service written in C++ and running in a scalable NCBI service mesh (linkerd, namerd, consul). When a CGV view is initially loaded, our gRPC service requests the alignment data needed for the particular page. On graphical pages, gRPC resolves an alignment identifier to a URL with prepared synteny/alignment data and the page loads the file at this URL into a WASM module which is written in C++ and compiled with Emscripten. The WASM module serves this data on demand to the page's JavaScript code which uses it for building the image and all user interactions.

In parallel, the list of assembly-assembly alignments and their metadata is sent to the selection menu (i.e., "Set up your view") on the CGV landing page. This allows the selection menu to report scientific and common species names, assembly accessions, and assembly names.

The alignment selection menu on the CGV landing page is a traditional web form page. We utilize the NCBI version of United States Web Design System (USWDS) design standards and components (https://www.ncbi.nlm.nih.gov/style-guide) to unify graphical design with other US government pages.

On the more graphically intensive ideogram and dotplot pages, graphical rendering is done in the user's web browser using WebGL using d3.js or pixi.js libraries, which allows for efficient interactivity, scalability, and fluidity of user interaction. Other elements of the page use jQuery/extJS and USWDS components. CGV reuses chromosome ideograms initially developed for NCBI's Genome Data Viewer [2].

Gene annotations shown in CGV are obtained from NCBI's public databases using NCBI Entrez Programming Utilities (E-utilities) (https://www.ncbi.nlm.nih.gov/books/NBK25501/). Annotation-build specific gene search is provided by an NCBI Datasets (https://www.ncbi.nlm.nih.gov/datasets/) gRPC service.

## Design of CGV application

A philosophy of user-centered design, which puts user needs at the forefront of decision-making, was an integral element in the development of the CGV. Participants for user research were recruited from members of the genomics research community who provided their contact information through feedback links on the CGV application and other sequence analysis tools at NCBI. Some of these researchers were previously familiar with CGV, while others had little or no experience with this tool. Data from user research testing sessions was compiled and analyzed for patterns in behavior, thereby allowing the team to validate that the design was moving in a direction that facilitated analysis of sequence alignment data. To date, we have conducted user research with over 30 different experts in the field of comparative genomics. We also evaluate the application for Section 508 compliance, which helps insures CGV performs well on mobile devices and is accessible to users with limited or no visibility.

## Supporting information

**S1 Appendix. Examples of CGV showing segmental duplications when non-best placed alignments are included in the view.** CGV's ideogram view displays best-placed alignments by default. Non-best placed alignments can be added from the "Adjust Your View" configure options (Fig 1F). (A) CGV view of whole-genome alignment of *Arabidopsis thaliana* and *A. suecica*, a polyploid hybrid of *A. thaliana* and *A. arenosa* [37]. The default view shows regions of similarity between *A. thaliana* and the *A. thaliana* subgenome of *A. suecica*. Including non-best placed alignments reveals additional alignment segments (marked by red boxes) that correspond to alignment between *A. thaliana* and the *A. arenosa* subgenome of *A. suecica*. https://www.ncbi.nlm.nih.gov/genome/cgv/browse/GCA_019202805.1/GCF_000001735.4/48965. (B–E) CGV alignments between human and chimpanzee (B, C, and E) and human and

bonobo (D) genome assemblies in regions that contain local segmental duplications. When non-best placed alignments are shown, additional alignment segments are displayed that identify potential additional copies of *BMPR2* (B), *EIF4A3* (C and D), or *EYS* related gene sequence (E). *EIF4A3* was previously reported to have multiple copies in both chimpanzee and bonobo relative to the human genome [38]. (B). https://www.ncbi.nlm.nih.gov/genome/cgv/browse/GCF_028858775.1/GCF_000001405.40/35595/0#NC_072400.1:104175446-105102154/NC_000002.12:202185329-203061800/size=1000,firstpass=0. (C). https://www.ncbi.nlm.nih.gov/genome/cgv/browse/GCF_028858775.1/GCF_000001405.40/35595/0#NC_072415.1:91293196-91791317/NC_000017.11:80086216-80514871/size=1000,firstpass=0. (D). https://www.ncbi.nlm.nih.gov/genome/cgv/browse/GCF_029289425.1/GCF_000001405.40/36375/9606#NC_073266.1:100957682-101607434/NC_000017.11:80031356-80681076/size=1000,firstpass=0. (E). https://www.ncbi.nlm.nih.gov/genome/cgv/browse/GCF_028858775.1/GCF_000001405.40/35595/0#NC_072404.1:68562328-78605127/NC_000006.12:62860419-76648142/size=1000,firstpass=0.
(PDF)

**S1 Data. Alignment coverage at different Mash distances for selected assembly pairs.**
(XLSX)

**S1 Fig. Merging, sorting, and ranking assembly-assembly alignments.** (A) Flowchart depicting how adjacent alignment segments are merged. Subsequently, alignments are split once again at large gaps. (B) Flowchart showing how overlapping alignments are separated, ranked, and re-merged. Reciprocal best-placed alignments are designated as "first pass," while the non-best placed alignment is designated "second pass."
(TIF)

# Acknowledgments

We thank Anne Ketter and Emily W Davis for help with project planning and coordination. We thank Guangfeng Song for aid with user experience research and analysis.

Thanks to Wayne Matten, Michelle Formica-Frizzi, and others in the NCBI customer service team for marketing, webinars, and video tutorials. Anatoliy Kuznetsov and Andrei Shkeda conducted early technical design work that influenced the architecture of the CGV application. Deanna M. Church and Mike DiCuccio participated in initial design and testing of the assembly alignment protocol. Finally, we thank members of the greater genomics research community who have participated in usability sessions and provided feedback and alignment requests to CGV. CGV was developed as a part of the National Institutes of Health's Comparative Genomics Resource (CGR) (https://www.ncbi.nlm.nih.gov/comparative-genomics-resource/).

# Author Contributions

**Conceptualization:** Sanjida H. Rangwala, Terence D. Murphy.

**Data curation:** Marina V. Omelchenko, Terence D. Murphy.

**Formal analysis:** Sanjida H. Rangwala, Dong-Ha Oh, Vamsi K. Kodali, Marina V. Omelchenko, Sofya Savkina, Joël Virothaisakun, Terence D. Murphy.

**Funding acquisition:** Kim D. Pruitt, Valerie A. Schneider.

**Methodology:** Dmitry V. Rudnev, Victor V. Ananiev, Dong-Ha Oh, Barrett Benica, Sofya Savkina, Joël Virothaisakun, Terence D. Murphy.

**Project administration:** Sanjida H. Rangwala, Valerie A. Schneider.

**Software:** Dmitry V. Rudnev, Victor V. Ananiev, Dong-Ha Oh, Andrea Asztalos, Evgeny A. Borodin, Nathan Bouk, Vladislav I. Evgeniev, Vamsi K. Kodali, Vadim Lotov, Eyal Mozes.

**Supervision:** Sanjida H. Rangwala, Dmitry V. Rudnev, Ekaterina Sukharnikov.

**Writing – original draft:** Sanjida H. Rangwala, Dmitry V. Rudnev, Dong-Ha Oh, Vamsi K. Kodali, Joël Virothaisakun, Valerie A. Schneider.

**Writing – review & editing:** Sanjida H. Rangwala, Dong-Ha Oh, Terence D. Murphy, Valerie A. Schneider.

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
