## [Editor Report · Decision Letter 0]

21 Oct 2023

Dear Dr Rangwala, 

Thank you for submitting your manuscript entitled "Interactive visualization of whole genome alignment between two eukaryotic genomes using NCBI’s Comparative Genome Viewer (CGV)" for consideration as a Methods and Resources Article by PLOS Biology. Please accept my apologies for the delay in getting back to you as we consulted with an academic editor about your submission.

Your manuscript has now been evaluated by the PLOS Biology editorial staff, as well as by an academic editor with relevant expertise, and I am writing to let you know that we would like to send your submission out for external peer review.

Once your full submission is complete, your paper will undergo a series of checks in preparation for peer review. After your manuscript has passed the checks it will be sent out for review. To provide the metadata for your submission, please Login to Editorial Manager (https://www.editorialmanager.com/pbiology) within two working days, i.e. by Oct 23 2023 11:59PM.

Kind regards,

Richard

Richard Hodge, PhD

rhodge@plos.org

PLOS

---

## [Decision Letter · Decision Letter 1]

15 Jan 2024

Dear Dr Rangwala,

Thank you for your patience while your manuscript "Interactive visualization of whole eukaryote genome alignments using NCBI’s Comparative Genome Viewer (CGV)" was peer-reviewed at PLOS Biology. It has now been evaluated by the PLOS Biology editors, an Academic Editor with relevant expertise, and by three independent reviewers. 

In light of the reviews, which you will find at the end of this email, we would like to invite you to revise the work to thoroughly address the reviewers' reports.

Reviewer #1 is very positive and has no requests. Reviewer #2 is fairly positive, and appreciates the effort as s/he has tried to build such a tool themselves. They question how well the CGV tool copes with comparisons that cross WGD events, and suggest that you either re-jig the algorithm to improve it, state explicitly that this is a limitation, or convince them that it *does* actually cope well. Because of this, they are sceptical that it would be as useful for plants (where WGDs are more common) as for animals; s/he also wonders how well it will perform where divergence is higher and synteny is less extensively conserved. Reviewer #3 is mostly positive, but wants you to do a direct comparison with related tools in order to make a case for the need for this new addition to the field; s/he also has a list of issues with the current performance and behaviour of the tool.

Given the extent of revision needed, we cannot make a decision about publication until we have seen the revised manuscript and your response to the reviewers' comments. Your revised manuscript is likely to be sent for further evaluation by all or a subset of the reviewers.

We expect to receive your revised manuscript within 2 months. Please email us (plosbiology@plos.org) if you have any questions or concerns, or would like to request an extension. 

**IMPORTANT - SUBMITTING YOUR REVISION**

*Re-submission Checklist*

*Published Peer Review*

*PLOS Data Policy*

Sincerely,

Roli Roberts

Roland G Roberts PhD

Senior Editor

PLOS Biology

rroberts@plos.org

on behalf of 

Richard Hodge, 

Senior Editor

PLOS Biology

rhodge@plos.org

REVIEWERS' COMMENTS:

Reviewer #1:

In this manuscript Rangwala et al. present a new tool for visualising whole genome comparisons using precomputed alignments of publicly available assemblies for hundreds of species across the tree of life.

They outline some widely used tools and visualisation methods for displaying genomic data and outline limitations with respect of assembly-assembly alignments. They show how the Comparative Genome Viewer presents alignment data in a clear and accessible manner on a platform (browser) open to all users, they do a step by step guide to using CGV and also highlight some interesting use cases.

They talk at length about front and back end set up and design choices and implementation for a good user experience, they also cover some usage limitations and potential new features to implement.

This is a nice well written paper, figures are clear and concise and relevant to the manuscript. There are no big concerns about the presented data and there is definitely space for a tool such as this.

Not something to address in the paper but two thirds of genomes in the alignment set are animals with 20% of the total being mammalian, this is not representative of the publicly available assemblies in GenBank. In addition, it would be worth having some assembly quality metric to help the end user to better understand the genome data, as the authors raise repeatedly the potential for there to be assembly errors.

I do recommend the paper for publication.

Reviewer #2:

Rangwala and colleagues have built a useful and user-friendly interactive visualization tool for pairwise comparison of genomic synteny and alignments. Having worked to build one of these myself, I commend the authors on the simplicity of the user-facing operations and the variety of downstream analytical tools. I have a few thoughts that I hope will improve the utility of the tool, or alternatively better contextualize the scale/scope of the results presented in the manuscript. 

It looks like the synteny finding algorithm doesn't deal well with whole-genome duplications. As far as I could tell, there is only one comparison available in the CGV that crosses a whole-genome duplication, that of Xenopus laevis vs. X. tropicalis (https://www.ncbi.nlm.nih.gov/genome/cgv/browse/GCF_000004195.4/GCF_017654675.1/38475/8355), and the alignments therein do not look good. Since the WGD is on the X. laevis branch (Fig. 2 here: https://www.nature.com/articles/nature19840), both homeologs should be similarly related to X. tropicalis, so primary and secondary alignments have no meaning with regard to the homeologs. Yet you can only see the duplication with the inclusion of secondary alignments. I understand that finding synteny from whole-genome alignments that cross a WGD is difficult (unlike with CDS, which is simple), but it is doable with the right context or algorithm. I'd strongly suggest either re-writing the algorithm to handle these cases, or state up front that the CGV cannot handle comparisons of variable ploidy. If you disagree with me and think that the GCV can handle WGDs, please include examples of simple comparisons, like Xenopus laevis  human, wheat  Brachypodium distachyon and Brassica napus (oilseed rape)  Arabidopsis thaliana, and show that genome coverage is equally good for all homeologs. 

Depending on the response to my above comment regarding WGDs, I think the statement that CGV is something that is useful for eukaryotes in general is not accurate. Really, it appears that it is only useful for single-copy genomes. I bring this point up because all of the plant species included in the CGV only extend to other genomes that coalesce before a whole-genome duplication event. However, in many cases, a user would want to use this tool in plants like they would in animals - to explore regions of interest between a genome and its related model system. Nearly all alignments from crops to the major plant models (Arabidopsis and Brachypodium) cross at least one WGD. There are ~5x more plant species than vertebrates … the CGV would not be useful for the vast majority of these.

The CGV is not going to be useful for comparisons of very diverged species that lack solid synteny. In amniotes, synteny is remarkably conserved and there still is ~50% genome coverage between Human and Xenopus: https://www.ncbi.nlm.nih.gov/genome/cgv/browse/GCF_000004195.4/GCF_000001405.40/46255/9606#NC_030677.2//size=10000 cover ~50% of the genome). However, as you go deeper (e.g. humans to zebrafish) or with less conserved genomes (https://www.ncbi.nlm.nih.gov/genome/cgv/browse/GCF_000002335.3/GCF_023101765.2/42855/7070), the utility breaks down. While this limitation is obvious to someone well-versed in comparative genomics, it won't be to the vast majority of users and needs to be stated up front.

It is stated that ~700 genomes are available - this makes it seem like one could compare all of these genomes to each other. This is not the case. The majority of the pre-computed alignments are to a very restricted set of genomes, mostly within the same species. I think this is fine, but this limitation needs to be stated clearly up front. Perhaps instead of having all genomes together in one dropdown, have a "project" or whatever selection that gets you to all genomes that can be compared? 

Genomes have variable naming conventions, and this can produce comparisons that look messy, but are actually very syntenic. The rat/mouse comparison illustrates this: there are some rearrangements, but these are hard to see since syntenic chromosomes are not stacked. The Phytozome comparative genomics viewer (https://phytozome-next.jgi.doe.gov/tools/synteny) handles this with the "sort scaffold" toggle. Incorporating this here would be helpful. 

Reviewer #3:

In this manuscript, the authors introduce the Comparative Genome Viewer (CGV), a web-based pairwise whole-genome alignment visualization tool hosted by the NCBI. The tool allows visualization of ~700 pre-computed pairwise alignments from eukaryotics species. The pairwise alignments are computed using BLAST, LASTZ, or imported from other sources and then post-processed using a custom algorithm. The visualization uses an existing "stacked linear browser" approach and includes gene annotations as tracks for both genome assemblies. The tool has links out to views within the existing Genome Data Viewer (GDV) for browsing within single genomes.

Overall, the CGV is an intuitive tool that provides access to a large number pairwise whole genome alignments. It should be useful to researchers needing to examine orthologous regions from previously published genome assemblies. It is not designed to handle newly-sequenced unpublished genomes, and thus generators of such data will need to use other tools for initial comparative analysis of their genomes. I have just one major critique of the manuscript with respect to its comparisons to existing tools and handful of minor comments.

Major comments:

1. There are several other large-scale providers of pairwise genome alignments and associated visualization tools. For example, the UCSC Genome Browser has a chain/net track that allows for visualizing pairwise alignments with another genome and can easily show rearranged segments (contradicting lines 67-70). The Ensembl website also hosts many pairwise alignments and has dedicated visualizations for viewing them. This manuscript should more strongly make the case for why an additional tool is needed in this space. For example, a figure showing how the these different tools display the alignment of the same region could help to show the strengths and weaknesses of these tools.

Minor comments:

2. Going to dotplot view from ideogram view puts you back at whole-genome level not the regions currently being viewed in ideogram view, which is counterintuitive.

3. The download button downloads entire whole-genome alignment, not the currently displayed part of the alignment.

4. It is unlcear how to download in XLSX format - this option was always greyed out even when zooming in to small regions. It is also unclear what information this format conveys.

5. Can other data tracks other than gene annotations be added?

6. I found the base level view interface difficult to understand, particulary in how it interacts with the main interface. Some further desciription or documentation of this interface would be helpful.

7. Figures in this manuscript need to be at higher resolution as many details are blurry - given that CGV can export in SVG this should be easily achievable.

8. A custom approach is described for processing initial alignments, however, no software is provided as an implementation. Ideally, this software should be made publicly available so that other researchers can evaluate the approach on other data.

9. Please cite and explain Mash distance at its first mention in the manuscript.

10. Line 60 - perhaps mention IGV as it is a popular desktop-based genome browser

11. An example figure showing a non-best alignment with one region in one genome alignment to multiple regions in another would be helpful to understand how this visualization works.

12. amylase family expansion example: it is unclear why there is an unaligned sequence segment in CHM13 if it contains additional copies of the alpha-amylase gene. Are these part of "next-best" alignments that are not shown? If not, why are they not part of an alignment?

---

## [Editor Report · Decision Letter 2]

13 Mar 2024

Dear Sanjida,

Thank you for your patience while we considered your revised manuscript "Interactive visualization of whole eukaryote genome alignments using NCBI’s Comparative Genome Viewer (CGV)" for publication as a Methods and Resources Article at PLOS Biology. This revised version of your manuscript has been evaluated by the PLOS Biology editors and the Academic Editor. Please accept my apologies, but I was unable to switch Dong-Ha to corresponding author for the submission as this is a permission that is only granted to the authors when the submission is sent back. I noted that your parental leave begins around 18th March so I hope that you still see this message, but I have cc'ed Dong-Ha in this decision letter to ensure that he is looped in to this correspondence. If you wish, Dong-Ha can be switched to corresponding author upon resubmission and then in principle switched back during the production process.

Based on our Academic Editor's assessment of your revision, I am pleased to say that we are likely to accept this manuscript for publication, provided you satisfactorily address the following data and other policy-related requests that I have provided below (A-E):

(A) We would like to suggest the following modification to the title:

“NCBI Comparative Genome Viewer (CGV) is an interactive visualization tool for the analysis of whole-genome eukaryotic alignments”

(B) You may be aware of the PLOS Data Policy, which requires that all data be made available without restriction: http://journals.plos.org/plosbiology/s/data-availability. For more information, please also see this editorial: http://dx.doi.org/10.1371/journal.pbio.1001797

-Supplementary files (e.g., excel). Please ensure that all data files are uploaded as 'Supporting Information' and are invariably referred to (in the manuscript, figure legends, and the Description field when uploading your files) using the following format verbatim: S1 Data, S2 Data, etc. Multiple panels of a single or even several figures can be included as multiple sheets in one excel file that is saved using exactly the following convention: S1_Data.xlsx (using an underscore).

-Deposition in a publicly available repository. Please also provide the accession code or a reviewer link so that we may view your data before publication. 

Regardless of the method selected, please ensure that you provide the individual numerical values that underlie the summary data displayed in the figure panels as they are essential for readers to assess your analysis and to reproduce it.

(C) Please also ensure that each of the relevant figure legends in your manuscript include information on *WHERE THE UNDERLYING DATA CAN BE FOUND*, and ensure your supplemental data file/s has a legend.

(D) Per journal policy, as the code that you have generated is important to support the conclusions of your manuscript, we require that you make it available without restrictions upon publication. Please ensure that the code is sufficiently well documented and reusable, and that your Data Statement in the Editorial Manager submission system accurately describes where your code can be found. 

(E) Please ensure that your Data Statement in the submission system accurately describes where your data can be found and is in final format, as it will be published as written there. 

We expect to receive your revised manuscript within two weeks. 

*Published Peer Review History*

*Press*

Kind regards

Richard

Richard Hodge, PhD

rhodge@plos.org

PLOS

---

## [Editor Report · Decision Letter 3]

8 Apr 2024

Dear Dong-Ha,

On behalf of my colleagues and the Academic Editor, Andreas Hejnol, I am pleased to say that we can accept your manuscript for publication, provided you address any remaining formatting and reporting issues. These will be detailed in an email you should receive within 2-3 business days from our colleagues in the journal operations team; no action is required from you until then. Please note that we will not be able to formally accept your manuscript and schedule it for publication until you have completed any requested changes.

PRESS

Best wishes, 

Richard

Richard Hodge, PhD

rhodge@plos.org

PLOS
